# Recent Developments in Biopolymer-Based Hydrogels for Tissue Engineering Applications

**DOI:** 10.3390/biom13020280

**Published:** 2023-02-02

**Authors:** Rikako Hama, Anudari Ulziibayar, James W. Reinhardt, Tatsuya Watanabe, John Kelly, Toshiharu Shinoka

**Affiliations:** 1Center for Regenerative Medicine, The Abigail Wexner Research Institute, Nationwide Children’s Hospital, 700 Children’s Drive, Columbus, OH 43205, USA; 2Department of Biotechnology and Life Science, Graduate School of Engineering, Tokyo University of Agriculture and Technology, 2-24-16 Naka-Cho, Koganei 184-8588, Japan; 3Department of Cardiothoracic Surgery, The Heart Center, Nationwide Children’s Hospital, 700 Children’s Drive, Columbus, OH 43205, USA; 4Department of Surgery, Cardiovascular Tissue Engineering Program, Ohio State University, Columbus, OH 43210, USA

**Keywords:** tissue engineering, hydrogel, drug delivery system (DDS), biodegradation, silk fibroin, collagen, elastin, chitosan, proteoglycan

## Abstract

Hydrogels are being investigated for their application in inducing the regeneration of various tissues, and suitable conditions for each tissue are becoming more apparent. Conditions such as the mechanical properties, degradation period, degradation mechanism, and cell affinity can be tailored by changing the molecular structure, especially in the case of polymers. Furthermore, many high-functional hydrogels with drug delivery systems (DDSs), in which drugs or bioactive substances are contained in controlled hydrogels, have been reported. This review focuses on the molecular design and function of biopolymer-based hydrogels and introduces recent developments in functional hydrogels for clinical applications.

## 1. Introduction

Research in regenerative medicine has made great strides in recent years. It aims to regenerate and repair tissues by enhancing the healing ability of organisms and increasing the ability of cells to proliferate and differentiate. It is useful for tissue repair that surpasses the self-healing ability by delivering several types of cells associated with tissue regeneration into the damaged area; however, simply supplying cells alone is not sufficient. Cells must interact with other cells and the extracellular matrix (ECM) to survive and function properly. Therefore, tissue engineering materials that provide cells with ECM-like surrounding environments for tissue regeneration are essential for the successful expression of biological functions of cells in the body. The development of applications that can restore native tissue-like structures and functions is a pressing challenge in tissue engineering. Several attempts have been made to elucidate the combination of cells, scaffolds, and bioactive factors and their interactions to provide a more optimal tissue repair environment [1]. Empirical research has revealed the requirements for optimal scaffold materials; these include the appropriate surrounding environment for tissue regeneration for each tissue and cell type. The material that served as a scaffold for cell accumulation, proliferation, and differentiation at the injury site is desirable to be gradually degraded and replaced by neotissue as tissue regeneration and maturation progresses. If the material is bioabsorbable, it does not need to be removed by surgical treatment to eliminate the restriction of the spatial regenerative environment. This is expected to improve the functionality of the resulting regenerated tissue, as it is useful for preventing chronic inflammation and acquiring an ordered hierarchical structure like native tissue. In this regard, biopolymer is bioabsorbable and enables the formation of a more effective healing environment, as it activates a wide range of cellular functions, not only in tissue regeneration but also in the degradation and absorption of materials. Material shape and constituent materials need to be suitable for scaffolds as mechanical supports, mechanical properties, and bioactive functions to promote stronger tissue regeneration. Among these, the use of forming methods for hydrogel as matrices that closely mimic the structure of the ECM is expanding and is being applied to the regeneration of various tissues and organs (from soft to hard tissues).

Hydrogels are being explored for a wide range of applications, particularly in soft tissues, owing to the diversity of their manufacturing and functionalization methods for controllable biodegradation and mechanical properties. The three-dimensional network framework of hydrogels can be easily adjusted to achieve the desired physical properties using physical parameters such as the degree of cross-linking, elastic modulus, and degradation rate as indicators [2,3]. Thus, shape and flexibility can conform to the target tissue, and the degradation period can be tuned to follow the progress of tissue regeneration (Figure 1). In addition, the ability to combine multiple functions and conjugate multiple bioactive molecules allows further bioactivity to be added, taking advantage of the biocompatibility of the material. These are attractive points that allow the design of materials in accordance with the established regeneration strategy for each tissue. The use of naturally derived polymers with biocompatibility and useful biofunctions as materials for hydrogels is gaining momentum [3]. In recent years, for the acquisition of even more highly matured regenerative tissues, the approach of functionalization in combination with drug delivery systems (DDS) has been frequently used, including simple methods of adding growth factors, biological signal factors (i.e., peptides), and drugs to 3D materials, as well as cell culture systems as scaffolds to which cells are added.

In this review, we focus on hydrogels based on biopolymers and summarize their usefulness and the recent expansion of research trends. Focusing on protein polymers, such as collagen and silk, and polysaccharide polymers, such as glycosaminoglycans and hyaluronic acid, which are often used as materials, the structure, biocompatibility, and biological activity have been reviewed. In addition, we explored the direction of high-functionalization methods based on these properties and summarized how biopolymer-based hydrogels could be studied as promising materials. 

## 2. Hydrogels

A hydrogel is a generic term for a 3D network structure composed primarily of polymers containing water as the solvent. This network structure is formed by cross-linking polymers via covalent and non-covalent interactions. Thus, biocompatible and biodegradable synthetic polymers, often natural polymers, are used as materials. A variety of chemical and physical cross-linking methods are available, and the material properties reflect the overall chemical structure of the monomers used (type of functional group, number of cross-linking points present, and molecular weight), composition and concentration of the raw material solution, cross-linking method, and ratio [4]. Moreover, the potential for the induction of tissue regeneration is influenced by biocompatibility, such as lower immune response, and hydrophilicity of the materials used. The hydrous capacity to retain water acts on the diffusion of molecules and cells held within the hydrogel through water movement associated with swelling in vivo and affects the degradation behavior in the long term. The use of protein- and polysaccharide-based natural polymers as components of native tissue, ECM has been widely explored. In addition to the basic biological advantages described above, natural polymers are often biodegradable and exhibit a range of low inflammatory and non-toxicity properties that make the materials and their degradation products suitable for inducing tissue regeneration. However, disadvantages based on complex and heterogeneous mesh structures remain a challenge for hydrogels of conventional synthetic polymers and natural polymers. One of the advantages of using hydrogels is their use in situ; however, gelation time and gel properties should be controlled with high precision [5]. The gelation point of hydrogels can be determined from rheological analysis of the sol-gel transition, as detailed by Yan et al. [6]. However, the exact kinetics of gelation and precise gelation of many natural proteins have not been elucidated. Therefore, for clinical use, it is necessary to overcome more precise control encompassing the rate of gelation/degradation reaction, regulation of the degree of cross-linking reaction, and gelation time suitable for in situ use. Especially the elastic modulus and swelling rate are more important, since they are also related to the sustained drug-release kinetics of DDS materials.

Many hybrid materials have been developed that take advantage of the diversity of cross-linking methods and material choices [7]. There are two main crosslinking methods that can be widely used or combined according to the physical properties of the target hydrogel or the chemical structure and biological activity of the selected material. Chemical methods of covalent crosslinking include the use of carbodiimide compounds (i.e., glutaraldehyde (GA), and 1-ethyl-3-(3-dimethylaminopropyl) carbodiimide/N-hydroxysuccinimide (EDC/NHS), radical polymerization, thermal gelation, enzymatic reactions, click chemistry, photo-oxidation, and radiation cross-linking. Physical methods that utilize changes in intermolecular interactions, which are often used with natural polymers, include self-assembly through hydrophobic interactions, van der Waals forces, hydrogen bonding, and electrostatic interactions [3]. In recent years, the use of “green chemistry” has gained attention because of concerns regarding the inflammatory properties caused by residual solvents and cross-linking agents. 

As for fabrication methods, in addition to the conventional method of creating a shape by gelation in a mold or after injection into a defective tissue area, 3D printing technology is being actively applied to fabricate more complex structures. For example, in hydrogels, it is not only possible to add cells or bioactive substances and print them but also to combine different polymer solutions for degradation or sustained release behavior purposes. The precise deposition of layers at the microscale by machine can also be used to fabricate complex shapes with exact dimensions according to a blueprint. Conventional 3D printing most commonly involves direct ink writing (DIW) or fused deposition modeling (FDM), in which ink is extruded [8]. The concept of 4D printing, the latest method, was proposed by Tibbits in 2013 as “3D plus times” as the fourth dimension [9]. This is obtained by designing everything from material structures to interaction mechanisms to suit the purpose so that the printed structure changes its morphology, properties, and function when subjected to external stimuli for an appropriate amount of time. As described in detail in the next section, stimulus-responsive hydrogels are responsive and suitable for 4D printing [10,11,12]. Material selection is discussed in the following sections, based on the structure and properties for application to various tissues.

## 3. Synthetic Polymers

Synthetic polymers such as poly(ethylene glycol) (PEG), poly(vinyl alcohol) (PVA), and poly(hydroxyethyl methacrylate) (poly-HEMA) are commonly used as hydrogels [13,14]. They are superior to naturally derived polymers in that they have a relatively homogeneous chemical composition; thus, the functional groups that confer hydrophilic and mechanical properties can be selected and designed. Biodegradable materials are important not only because they do not require additional procedures for removal but also because they allow for the replacement of the host’s neoplastic autologous tissue as it degrades over time. Therefore, materials should be designed to match the degradation mechanism and rate with the healing environment of the target tissue. Factors that influence degradation include molecular weight, copolymer cost, and the number of branches of the monomer substituents. To evaluate the toxicity of a material, the entire process, from the chemical reaction to cross-linking, should be considered: residual monomers, solvents, and by-products, such as hydrolysis or enzymatic degradation products, should be stable and not cause excessive and chronic immune reactions. Furthermore, degradation products should be eliminated from the body without long-term accumulation.

Polyesters are a class of biodegradable synthetic materials with ester bonds in the main chain, including poly(glycolide) (PGA), poly(lactide) (PLA), poly(caprolactone) (PCL), and poly(trimethylene) (PTMC) [13]. Polyesters are used in the form of sponges and nanofibers as tissue engineering materials, especially in cardiovascular [15]. In vivo, ester bonds undergo enzymatic and nonenzymatic hydrolysis, and the entire polymer is degraded by bulk erosion or erosion. The degradation products include lactic acid, glycolic acid, or caprolactone. In addition, many polyesters, with the exception of PLA and PCL, are hydrophobic and are often combined with other hydrophilic materials; Gordon et al. incorporated aspartic acid and PEG to obtain hydrogels for cell encapsulation by free radical polymerization [16]. Human adipose-derived stromal cells were encapsulated in the cross-linked gel at compressibility comparable to that of soft tissue and maintained >80% viability even after seven days. Several composite materials also appear in the sections that follow. These include light-curing and heat, pH, and other stimuli-responsive polymers, the use of which has increased in recent years. In situ gel formation in defects is helpful in adapting to wound sites of different shapes and sizes from patient to patient [17]. In addition, by designing control over the disintegration and degradation behavior of the hydrogel, it is expected to act as a precise DDS base. For example, N-isopropyl acrylamide (NIPAM), which has a lower critical solution temperature (LCST) of about 32 °C, close to the in vivo temperature, has been actively applicated for tissue engineering among temperature-sensitive polymers [18]. Nistor et al. have developed a super-porous hydrogel, a composite of sponge-hydrogels, by impregnating a collagen sponge with NIPAM and then radical polymerizing it [19,20]. In this composite hydrogel, protection of collagen from enzymatic degradation and suppression of swelling ratio suitable for use in bone materials were achieved, and it showed no inhibition of cell proliferation. PEG-polyester copolymers also attract attention for their prospects as thermosensitive and biodegradable hydrogels. In particular, hydrophilic PEGs have excellent biocompatibility and have been used extensively in the medical field. Moreover, PEG-polyester copolymers and PEG-polypeptide copolymers have overcome non-biodegradability and rapid dissolution by introducing hydrophobic segments, and derivatives are being actively developed [21]. The trend is that they are mainly used as copolymers with other synthetic polymers, especially PCL and poly (PLGA). The current hotspot is developing base material for DDS by adding drugs and bioactive factors for cell encapsulation and scaffold materials rather than complexing with biopolymers. Fu et al. have developed PEG-PCL-PEG/collagen/nanohydroxyapatite (n-HA) as an injectable hydrogel for bone regeneration [22]. Based on previous studies [23], collagen was added to PEG-PCL-PEG/n-HA hydrogels to improve the performance of induced bone regeneration. In a cranium defects model in rabbits, accelerated new bone formation was demonstrated. This may be due to the exposure of collagen, which degrades more slowly than PEG-PCL-PEG, thereby modulating cell proliferation and function. Thus, strategies to induce tissue regeneration by combining multiple biopolymer and bioactive factors are expected to increase. To accelerate such stimuli-responsive polymer and biopolymer composites in the future, it is expected to accumulate knowledge on comprehensive and long-term tissue regeneration behavior in vivo.

There are several review articles on synthetic polymeric tissue engineering materials that will not be discussed in detail in this review [7,8,24,25]. It is emphasized here that low cell affinity often makes hydrogels unsuitable for stand-alone use, especially for inducing tissue regeneration or cell culture substrates.

## 4. Natural Polymers, Protein Polymers

Regenerative medicine products, especially those used at biological interfaces that come in contact with biological components such as living tissue or body fluids (for example, scaffold materials), must be carefully evaluated for biocompatibility, including components and their degradation products [26]. Biocompatibility, that is, not being recognized as a foreign substance by the living body is essential and is selected based on its functions. In other words, in addition to the appropriate shape, mechanical properties, and hydrophilicity to support tissue regeneration, biodegradable ability according to the target tissue or regeneration strategy is important. The inherent biocompatibility and biological activity of natural polymers make them promising candidates for biomaterials that functionally mimic the composition and structure of ideal ECMs. Moreover, the degree of compatibility and the mechanism of expression are often empirical and unresolved owing to the diversity of the biological environment. In natural polymers derived from other animal species, it is sometimes important to avoid or reduce toxicity due to foreign body reactions caused by the origin of the material. However, they are still being studied in increasing numbers owing to their abundant cytophilicity, nontoxicity, hydrophilicity, and availability, and many are already commercially available. This section covers some typically used protein-based polymers.

### 4.1. Collagen, Gelatin

Collagen is the major protein component of connective tissue and basement membranes and provides mechanical support that confers resistance to plastic deformation. Approximately 30 genetically distinct collagens exist in numerous types, with different tissue distributions and mechanical properties. Collagen type 1 is the most abundant protein. The primary structure of collagen is a -(Gly-X-Y-)n- (frequently, X = Proline, Y = hydroxyproline) tripeptide [27]. Most form a triple helical structure, and the presence of the less bulky Gly contributes to the formation of the helical structure. The helical region, consisting of approximately 1000 polypeptide chains, forms collagen fibers via self-assembly [28]. Collagen was used in tissue engineering because of its biocompatibility and biodegradability, such as low antigenicity and low inflammation, as well as its mechanical properties. Recently, collagen has been increasingly used for its biological activities, such as its cell adhesion sites, ability to alter cellular differentiation, and role in regulating angiogenesis [29]. Collagen derived from other animal sources, primarily pig skin and rat tails, was used in this study.

There are many collagen-based hydrogels; however, damage during the extraction process from living organisms compromises the original mechanical properties and stability [30]. Thus, modification of the mechanical strength is often required. This is a common concern for natural polymers, which is also related to their rapid degradation rate. Their role as engineered or mechanical supports that provide scaffolds for cells to migrate or add is limited by their degradation rates. Therefore, cross-linking is often added during the material processing stage [31]. The cross-linking method frequently used with collagen is chemical cross-linking using cross-linking agents, such as GA or EDC/NHS [32]. It is possible to further improve thermal stability by linking collagen chains or by linking collagen to other polymers, but there are concerns about residual toxicity. Physical cross-linking methods, such as UV light treatment and dehydration heat treatment, can avoid these toxic effects but are susceptible to protein conformational breakdown. In addition, enzymatic cross-linking using genipin, lysyl oxidase, or transglutaminases is attracting attention because it can be used not only to modify physical properties through specific cross-linking but also to incorporate functionalized proteins [33,34,35].

Therefore, gelatin obtained by the hydrolysis of collagen is also used. Gelatin is a single or multichain polypeptide that is attractive because of its low immunogenicity, high solubility, and complete absorption into organisms [36,37]. Nonwoven gelatin hydrogel fabrics are used as cell culture substrates that can be uniformly distributed in the interior. Application to a mouse model of full-layer skin defects showed material degradation, followed by replacement with ECM, infiltration of mesenchymal stromal cells, and abundant vascularization [38]. However, the lower degree of tissue regeneration induced by inflammatory reactions compared to the long-used pig collagen sponge material suggests a challenge in using gelatin, which degrades faster than collagen, as a scaffold. In addition, various cross-linking procedures and combinations with other materials, such as collagen, have been used to resolve low thermal stability and mechanical properties [39]. Gelatin methacryloyl (GelMA), obtained by methacrylation of gelatin, allows photo-cross-linking under mild room temperature and neutral pH conditions [40,41]. Nazir et al. evaluated the wound-healing effects of GelMA hydrogels containing microcrystalline cellulose in a rat skin model of full-layer defects [42]. Compared to the GelMA hydrogel, the addition of the cellulose showed faster wound closure and keratinocyte development in the neoplastic epithelium.

### 4.2. Elastin

Elastin is a protein that makes up the connective tissue ECM and provides elasticity to tissues. Tropoelastin, a water-soluble precursor, becomes insoluble elastin upon cross-linking [28]. Elastin is actively used in skin substitutes and vascular grafts owing to its biocompatibility, biodegradability, and elasticity. However, it is used less frequently as a material for hydrogels because of its potential for contamination during the extraction process from elastic fibers and its insolubility in water. Often used in fusible form or composited to provide elasticity to other materials, such as tropoelastin, elastin-like polypeptides (ELP) [43,44], and genetically engineered elastin. The cross-linking methods of choice include cross-linking agents such as GA, click chemistry with additional functional groups such as azide groups, and enzymatic cross-linking, such as genipin [45,46]. The combination of encapsulated cells and hydrogels has potential applications in tissue repair, especially for cartilage tissue, which has a low regenerative capacity, but the mechanical characteristics and low adhesion strength are challenges. ELPs derived from human tropoelastin gene sequences were cross-linked and implanted in a rabbit model of a full-layer cartilage defect of the femoral condyle [46]. Even after six weeks, the material was stable and visible in the defect area, with no increase in the local inflammatory response. In addition, the formation of new bone around the material was accelerated, and integration with normal tissue through adjacent cartilage tissue was achieved. The hyaluronic acid/ELP hydrogel of Sani et al. utilizes rapid in situ photocrosslinking to develop an adhesive hydrogel for use in cartilage (Figure 2) [47]. Next, 2% methacrylated hyaluronic acid (MeHA)/10% ELP exhibited the highest burst pressure (19.87 ± 6.92 kPa), which surprisingly was about 13 times higher than commercial tissue adhesives (Evicel and Coseal). Furthermore, the addition of zinc oxide (ZnO) nanoparticles to provide antimicrobial properties did not compromise the mechanical properties of the gel, and antibacterial properties, diffusion, and growth of hMSCs in vitro were demonstrated. MeHA/ELP hydrogels were implanted subcutaneously in rats and showed a weight change of about 40% after four weeks, but about 20% after eight weeks. This was considered to be tissue infiltration associated with gel degradation, based on the fact that H&E staining confirmed autologous tissue replacement and hydrogel biodegradation. Regarding the degree of inflammatory cell mobilization around the hydrogel, no lymphocyte (CD3) infiltration was observed, and macrophage (CD68) showed a minor infiltration after four days but disappeared by four weeks. Therefore, it is expected to be applied as a tissue engineering material with high biocompatibility that does not induce persistent inflammatory reactions, antibacterial properties, and high adhesive properties. Soucy et al. created a fish-derived GelMA/tropoelastin composite hydrogel with neuroregeneration as an alternative to fibrin-based adhesives [39]. In vitro, it supported the proliferation of Schwann cells and the proliferation and spreading of encapsulated glia. In rat subcutaneous tissue, the gel was completely degraded in approximately eight weeks and showed autologous tissue replacement with low inflammation and cellular infiltration. 

### 4.3. Fibrin

Fibrin is a protein involved in hemostasis via the coagulation cascade during tissue repair in living organisms. Usually present in the blood in an inactive form called fibrinogen, fibrin is activated during hemostasis due to tissue injury, which forms an extensive fiber network. Therefore, fibrin is an excellent source of material because it can be prepared from autologous plasma. Therefore, fibrin glue has been mainly used as a hemostatic agent in surgical procedures, as a scaffold for cell adhesion in the primary cell and stem cell cultures, and as a coating for substrate surfaces [48]. Fibrin hydrogel is made from purified fibrinogen and thrombin. It has been used to promote angiogenesis and neurite outgrowth and has also been extensively studied in cardiovascular, ocular, and cartilage applications [49,50]. However, due to the challenges of gel shrinkage, low mechanical strength, and rapid degradation, composite applications have been deployed more often than stand-alone applications. Here, there are a number of combinations with other polymers. For example, a combination with PEG has improved the stiffness of hydrogels as structural supports [51]. Due to its ability to induce regeneration and biocompatibility, fibrin is frequently used in the 3D printing field, but its difficulty in handling it as a bio-ink is a challenge. Fibrinogen is difficult to maintain its shape, and with fibrin, it is necessary to overcome technical challenges such as ink extrusion due to the high viscosity of the solution [23]. Hyaluronic acid and gelatin are typically incorporated in the form of polymer blends to improve printability and regulate biological properties. Zhang et al. fabricated 3D-printed urethral materials using two types of solutions: PCL/PLCL (50:50) solution and fibrin gel solution containing cells [52]. Printing itself did not decrease cell viability, but after seven days of printing, it decreased by about 10%. This could be due to fewer nutrients and oxygen being supplied to the cells. When culturing cells in a hydrogel, it is necessary to combine the use of a dynamic perfusion bioreactor (Table 1). 

### 4.4. Silk, Silk Fibroin

Silk is a fiber protein produced by arthropods such as silkworms, bees, and spiders [67,68,69]. A complex of multiple proteins, silkworms, spiders, hornets, and bee silk are being used within the clothing and medical fields because of their superior mechanical strength and biocompatibility. The most widely used silk is silk fibroin (SF) produced by *Bombyx mori*. SF is the main ingredient of silkworm cocoons and has a long history of use as a surgical suture. Recently, SERI^®^, an absorbable mesh for soft tissue reinforcement, was approved by the US Food and Drug Administration (FDA) [70]. Despite its insect origin, it has been actively used in tissue engineering because of its biocompatibility, mild biodegradability, low inflammatory properties, promotion of fibroblast migration and ECM expression, wound healing, rigid mechanical properties, and thermal stability [71,72]. SF is a hydrophobic crystalline protein responsible for the strength and stiffness of silk fibers. The highly aggregated β-sheet crystal structure, formed mainly by hydrophobic sequences of (Gly-Ala-Gly-Ala-Gly-Ser)_n_, are scattered in amorphous portions constituted of sequences with polar and bulky side chains such as tyrosine residues [73]. SF can be dissolved in aqueous solutions of lithium bromide(LiBr), calcium chloride(CaCl_2_), etc. and can be easily processed into various forms such as films, nanofibers, sponges, and hydrogels [69,74,75,76,77]. Due to its mechanical properties and biodegradability can be widely controlled by adjusting the conditions for induction of recrystallization, SF has been extensively studied in soft tissues such as skin and blood vessels, as well as in hard tissues such as bone. However, unlike other animals or insect-derived proteins, SF does not have a cell adhesion domain, so functionalization by genetically modified silkworms [78], chemical modification [79], or material blending [80,81] is frequently used. Physical cross-linking methods are actively used for fabricating SF hydrogels: such as heat, vortexing, and sonication, and high shear forces by extrusion of highly concentrated solutions [74,82]. More moderate conditions such as lowering of pH, the addition of alcohol, genipin, and enzymatic cross-linking are also used [82]. Kambe et al. developed SF hydrogels by incorporating functional peptides using physical cross-linking methods to induce angiogenesis [54]. Angiogenesis-inducing peptides include cell-adhesive Arg-Glu-Asp-Val (REDVs), vascular endothelial growth factor mimicking peptides (QK peptides), matrix metalloprotease (MMP)-cleavable, and hydrophobic sequence in the formation region of SF β-sheet crystals. Subcutaneous implantation in rats suggested that the QK peptide, released from the SF hydrogel by MMPs secreted during the healing process, acts to recruit endothelial cells (ECs) into the gel. The group containing this fusion peptide produced highly efficient angiogenesis between four and eight weeks, approximately twice that of the group without the peptide. Dali et al. developed an in situ hydrogel for the treatment of dry eye by blocking fluid tear drainage [64]. Degradable hydrogels matching the irregular shape of the tear ducts were formed by visible light cross-linking. Indocyanine green fluorescent tracer nanoparticles (FTN) were incorporated in a network of methacrylate-modified SF (SFMA) (Figure 3). SFMA or SFMA/FTN hydrogels were injected subcutaneously into mice to gelate, and a mild inflammatory response peaked after seven days and persisted until eight weeks. Furthermore, when the hydrogels were formed into the lacrimal ducts using a rabbit dry eye model, lacrimal fluid retention was greatly improved. Fluorescence signal detection of the SFMA/FTN hydrogels showed that the gels remained in the tear ducts, indicating the release of the FTN tracer based on gel degradation. Interestingly, tissue sections of the entire tear duct showed no evidence of inflammatory cell buildup around the duct wall and no scar tissue formation. This material is expected to be useful as a next-generation material with a lower risk of complications than existing silicone tear plugs. 

## 5. Natural Polymers, Polysaccharides

Polysaccharides are receiving increasing attention for their potential usefulness as biomaterials. This is because of the potentially important functions of sugars in the field of immunology and technological developments, such as the artificial synthesis of oligosaccharides, as described by Suh et al. [83]. One of the properties of polysaccharides is that they form hydrogels depending on the structure of their constituent sugars and substituents. These can be divided into two major categories: hydrogen-bonded gels, such as agarose and chitosan, which gel with heat or pH, and ionic types, such as alginates.

### 5.1. Proteoglycan

Proteoglycans are macromolecules that make up the ECM of articular cartilage and other tissues and are composed of glycosaminoglycan (GAG) chains covalently attached to a core protein. GAGs are long linear heteropolysaccharides consisting of repeating disaccharide units of uronic acid and amino sugars containing sulfate or carboxyl groups. GAGs exhibit a high charge density and hydrophilicity owing to their low crystallinity. Owing to their low molecular weight and lack of gel-forming ability, they are often combined with other charged polymers, especially chitosan, a cationic polysaccharide [83].

Hyaluronic acid is another type of GAG present in the ECM of mammalian cartilage and skin, where it acts as a lubricant to retain water. It is composed of glucuronic acid and N-acetyl-glucosamine and can have a high molecular weight. It is also involved in wound healing processes, such as cell signaling, promotion of cell migration, angiogenesis, and ECM organization, is completely resorbed through multiple metabolic pathways, and shows improved mechanical properties depending on the degree of methacrylation [84,85]. Advances in microbial fermentation technology are expected to expand industrial applications through production systems to avoid the risk of contamination from animal sources. Hydrogels of MeHA/ELP [47] and HA/dopamine-coated graphene oxide by oxidative coupling [58] have also been reported.

### 5.2. Chitin, Chitosan

Chitin is one of the most abundant organic substances, and poly-N-acetyl-glucosamine is a natural amino polysaccharide. It is the second most biosynthesized substance in nature after cellulose and is an underutilized resource with potential for future engineering and medical applications. It is a major component of the exoskeletons of crustaceans and insects, and the cell walls of fungi [86]. Chitosan, used as a biomaterial, can be obtained from the shells of crustaceans such as shrimp and crabs by alkaline deacetylation, such as sodium hydroxide, and is recovered as poly N-acetyl-glucosaminoglycan [87]. Chitosan is a linear polysaccharide consisting of (1-4)-D-glucosamine and n-acetyl-D-glucosamine linked by a beta(1-4)glycosidic bond [28]. Chitosan exhibits biocompatibility, biodegradability, antibacterial and anti-inflammatory properties, and wound healing ability, and is similar in structure to GAG in ECM [86,88,89,90,91]. The degree of deacetylation treatment and molecular weight obtained allow control of the mechanical properties and degradability [86,88]. In addition, although the degradation products are non-toxic and non-immunogenic, and are increasingly attracting attention, especially for use in the medical field, the scarcity of methods for processing chitosan forms is a challenge [83]. Chitosan can be dissolved in dilute acidic solutions to obtain hydrogels with cell affinity. Chitosan/PEG hydrogels with intramolecular reversible hydrogen bonding and self-healing properties have been prepared for healing ulcerative skin wounds [59]. In addition to antimicrobial activity against *Staphylococcus aureus* and suppression of inflammation-induced cytokine levels derived from chitosan, chitosan was found to promote the migration of human dermal fibroblasts (HDF) in vitro. In addition, ex vivo protection against the degradation of elastic collagen fibers was obtained in a human ulcer model. Combinations of hair-derived keratin/chitosan [55], carboxymethyl chitosan (CMCS)/PVA [60], chitosan/antimiRNA-138 nanoparticles [64], and chitosan/silk fibroin/tannic acid/iron ions [61] have been applied to hydrogels. Additionally, in bone tissue engineering applications, chitosan is often composited with inorganic materials as a base material. This is a useful composite strategy since natural bone is composed of collagen and hydroxyapatite, an inorganic component. Polyhedral oligomeric silsesquioxanes (POSS) have a hybrid composition of inorganic SiO_2_ and organic silicone polymers, and the organosilicone polymer, provides stable covalent and/or noncovalent bonds with the biopolymer, resulting in improved mechanical properties [65,66].

### 5.3. Alginate

Alginate is a polysaccharide component of brown algae cell walls and bacterial capsules and is a natural anionic polymer. Commercial alginate is mainly extracted from brown algae (*Phaeophyceae*) by alkaline treatment with NaOH [92]. Alginate is obtained by dilute hydrochloric acid treatment of the alginate and purification to obtain water-soluble sodium alginate powder. The chemical structure is (1,4)-linked β-d-mannuronate and α-l-guluronate, which are linear copolymers organized in a block-like structure. Sometimes extracted from multiple brown algae, the mechanical properties of alginate-based hydrogels vary according to differences in sugar content/ratio, arrangement, and block length [93]. Thus, depending on the chemical structure and purity of the alginate from which it is derived, there is a range in the physical properties and degree of biocompatibility of the resulting hydrogels. In general, they are non-inflammatory and non-toxic, but this can be a challenge in terms of cell adhesion and low in vivo degradability. Functionalization by chemical bonding is used to modify cell affinity and mechanical properties of hydrogels [94]. The promise of DDS functionality through self-assembly led to the preparation of amphiphilic alginate derivatives by incorporation of hydrophobic functional groups. Acylated sodium alginate-g-polytetrahydrofuran graft copolymers(ASA-g-PTHF) by a combination of living cationic ring-opening polymerization and a grafting-onto method [95]. Segmentation of the main and side chains by microlayer separation enabled a significant improvement of the water contact angle of the polymer surface. Furthermore, ibuprofen-loaded microspheres showed pH sensitivity and low adsorption of bovine serum albumin. Based hydrogels containing exosomes derived from adipose-derived stem cells (ADSCs) for tissue repair and especially scar suppression [62]. Exosome secretion from the hydrogel showed highly efficient sustained release in the initial 72 h, followed by continued release with gel degradation over time. A rat skin model of total skin defects showed improvement angiogenesis and collagen deposition in the exosome-containing hydrogel group.

## 6. Conclusions

Despite a great deal of research into the structure and composition of hydrogels and their potential for inducing regeneration in all types of tissues, the precise control of their function and their potential are not fully understood. The hydrogel products commercially available as tissue engineering materials, such as DDS systems and wound dressings, are limited. As medicine becomes more sophisticated, hydrogels, especially those that can be used in situ, have the potential to be a breakthrough material in that they are more adaptable to individual patients and have multiple functions. Hydrogels, including biopolymers, are attractive for their biocompatibility and biodegradability and their ability to modulate cellular functions essential for tissue regeneration, such as proliferation and differentiation. Years of research have led to selecting other more effective polymers and bioactive substances based on their chemical structure-based properties rather than simply combining materials to improve their functions. Furthermore, with the progress in the development of synthetic polymers, stimulus-responsive and mechanical properties are being acquired by introducing their molecular design. Not only as conventional capsule substrates for cells and drugs but also in tissues ranging from skin and blood vessels to hard tissues of bone, their applications are expanding. Longer-term, more disease-like animal models and comprehensive evaluation of the relationship between hydrogel degradation, DDS behavior, and tissue regeneration processes are expected to shorten the time to reach the clinical application.

## Figures and Tables

**Figure 1 biomolecules-13-00280-f001:**
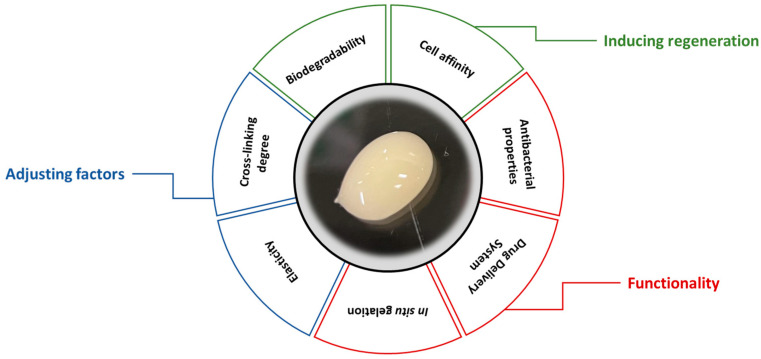
Requirements for ideal hydrogels.

**Figure 2 biomolecules-13-00280-f002:**
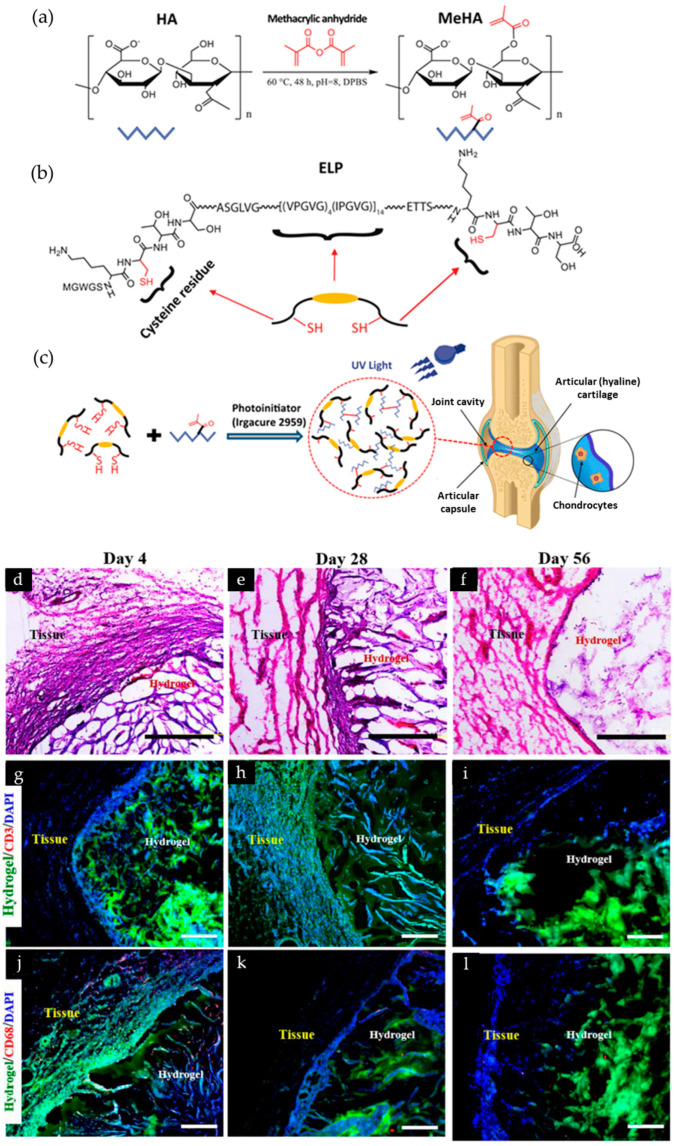
Schematic of methacrylated hyaluronic acid(MeHA)/elastin-like polypeptides(ELP) chemical structure. (**a**) HA methacrylation to form MeHA; (**b**) chemical structure of ELP; (**c**) schematic diagrams of photo-cross-linking of MeHA/ELP hydrogels and potential application as the adhesive composite. (**d**–**i**) In vivo biocompatibility and biodegradation of MeHA/ELP hybrid hydrogels in a rat subcutaneous implantation model. (**d**–**f**) Hematoxylin and eosin staining of MeHA/ELP sections (scale bars = 500 μm); (**g**–**i**) fluorescent images at 4, 28, and 56 days of immunohistofluorescent analysis (scale bars = 200 μm). Green, red and blue colors in (**g**–**l**) represent the MeHA/ELP autofluorescent hydrogels, the immune cells, and cell nuclei (DAPI), respectively. Hydrogels were formed by using 2% MeHA and 10% ELP at 120 s UV exposure time. Adapted with permission from [47]. Copyright 2018 American Chemical Society.

**Figure 3 biomolecules-13-00280-f003:**
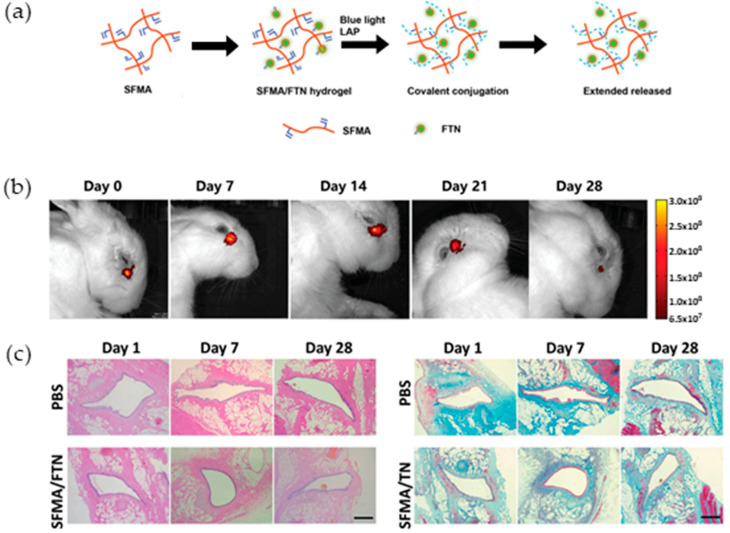
Synthesis and application of methacrylate-modified silk fibroin(SFMA)/fluorescent tracer nanoparticles (FTN) hydrogel plugs. (**a**) Schematic of the release of FTN from the SFMA hydrogel; (**b**) Long-term monitoring of SFMA/FTN hydrogel localized in the lacrimal sac of dry eye rabbits; (**c**) H&E and Masson staining observations for the biocompatibility assay after 1, 7, and 28 days of PBS and SFMA/FTN hydrogel in situ injection at the lacrimal duct, respectively (scale bars: 200 µm). Images reproduced with permission [64] ©WILEY-VCH Verlag GmbH & Co. KGaA, Weinheim.

**Table 1 biomolecules-13-00280-t001:** Natural polymer-based hydrogels.

Purpose	Materials	Cross-Linking Methods	Released Substrates	Functional Evaluation	Ref.
Antibacterial material	Methacrylated hyaluronic acid(MeHA)/elastin-like polypeptides(ELP)	Photopolymerization	Zinc oxide(ZnO)	in vivo (Subcutaneous implantation in rats)	[47]
Cell encapsulation	Asp-containing polyethylene glycol(PEG)	Graft polymerization (Potassium persulfate, tetramethylethylenediamine)	Human adipose-derived stromal cells	in vitro	[16]
	Gelatin methacryloyl(GelMA)	Photopolymerization	NIH3T3 fibroblasts	in vitro	[40]
	Collagen/MeHA	Photopolymerization	NIH3T3 fibroblasts	in vitro	[53]
Angiogenesis	Fibrin	Plasma-derived factor XIII	None	in vitro	[49]
	Silk fibroin(SF)	Heat treatment	Vascular-inducing peptide	in vivo (Subcutaneous implantation in rats)	[54]
Tissue regeneration	Keratin/chitosan	Crosslinker (Sodium tripolyphosphate )	None	in vitro	[55]
Vascular	PEG-diacrylate/fibrin	Photopolymerization	None	in vivo (Subcutaneous implantation in rats)	[51]
	ELP conjugated with C-peptide	Heat treatment	None	In vivo (Diabetic model in mice)	[56]
Skin	GelMA	Photopolymerization	6-deoxy-aminocellulose	in vivo (Wound healing model in rats)	[57]
	Hyaluronic acid/ dopamine/Graphene oxide	H_2_O_2_/HRP catalytic system	None	in vivo (Skin defects model in mice)	[58]
	Chitosan(CHI)/PEG	Crosslinker(NHS/EDC)	Antibiotics, anti-inflammatory drug(NSAID)	in vivo (Acute systemic toxicity assay in mice) ex vivo (Wound healing study in human ulceration model)	[59]
	Carboxymethyl chitosan (CMCS)/poly(vinyl alcohol)	Physical freeze-thaw cycling method	bFGF-loaded alginate microspheres	in vivo (Burn wound model in rats)	[60]
	Chitosan/silk fibroin	Photothermal method	Annic acid/ferric ion(TA/Fe3+)	in vivo (Skin defects model in mice)	[61]
	Alginate	Gelation with divalent cations	Exosomes from adipose-derived stem cells	in vivo (Skin defects model in rats)	[62]
Eye	Methacrylated silk fibroin (SFMA)	Photopolymerization	Fluorescence tracer nanoparticle(FTN)	in vivo (Dry eye model in rabbits)	[63]
Cartilage	Elastin-like polypeptides(ELP)	Genipin cross-linked	None	in vivo (Osteochondral knee defect model in rabbits)	[46]
Bone	Chitosan/β-sodium glycerol phosphate(CS/GP)	Heat treatment	Stromal cell-derived factor-1α, chitosan/tripolyphosphate/ hyaluronic acid/ antimiRNA-138 nanoparticles	in vivo (Cranial defect model in rats)	[64]
	Chitosan/polyhedral oligosilsesquioxane(POSS)	Genipin cross-linked	Ketoprofen,	in vitro	[65,66]
Urethral	PCL/PLCL (3D printed scaffold)	Heat treatment	Encapsulated urothelial cells(UCs) and smooth muscle cells(SMCs) in fibrin/Gelatin/HA	in vitro	[52]

## Data Availability

Not applicable.

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
