# Peer review of "Recent Developments in Biopolymer-Based Hydrogels for Tissue Engineering Applications"

_biomolecules, 2023, doi:10.3390/biom13020280_

Round 1

Reviewer 1 Report

The current review overviews different hydrogel systems for tissue engineering system. However, I feel this review require a significant update over important aspects -

1) Synthetic polymers like PEG and its derivatives have significantly been used in tissue engineering. Its important that the authors make a separate section on such polymers and provide details over it.

2) The current market has moved towards the implication of stimuli responsive hydrogels and their utility in 4D printing. In order to improve the readership of the article, the authors must comprehensively include section on the same.

3) Expanding the table to add more information would be valuable.

4) Incorporating or evaluating the compatibility of currently available hydrogel systems with current high end fabrication technologies must be added to expand the scope of the article.

The figure quality must be improved. In figure 1, was hydrogel image taken from somewhere. If yes, kindly add permissions. Else ignore. 

Author Response

Thank you so much for your kind and constructive comments. Please see the attachment.

Reviewer 2 Report

Dear Authors,

please find   my suggestions below

Kind regards

The manuscript describes the molecular design and function of biopolymer-based hydrogels and introduces recent developments of functional hydrogels for clinical applications. In my opinion, the manuscript is really interesting. Consequently, the manuscript can be accepted for publication in this journal after some revisions.

1.       It is clear that the papers chosen and reviewed for the topic in question depend on the discretion of the authors, just as I realise that authors cannot cite all articles related to the topic. However, I believe that articles from the last two years such as Materials 2022, 15(22), 8208; https://doi.org/10.3390/ma15228208, Int J Mol Sci. 2020 May; 21(10): 3442, doi: 10.3390/ijms21103442 RSC Adv., 2020, 10, 11325-11334 DOI: 10.1039/D0RA01636E, International Journal of Biological Macromolecules 142, 2020, 643-657, https://doi.org/10.1016/j.ijbiomac.2019.10.006, Macromol. Chem. Phys. 2022, 223, 2100366, https://doi.org/10.1002/macp.202100366 and Journal of the Mechanical Behavior of Biomedical Materials 125, 2022, 104966, https://doi.org/10.1016/j.jmbbm.2021.104966.and so on, are to be discussed, I noted that the authors refer to papers from 1985, 1993, 1995, 1998 and so on.

2.       Please leave a space between the sentence and the bibliographic reference throughout the text.

3.       Line 74 there is an error

4.       Figure 2 is not formatted well, please fix it

5.       The caption, line 294, 'Table 1. Hydrogels based on natural polymers', should be listed before the table

6.       Lines 295-298 should go up at the end of paragraph 4.3

7.       Conclusions are too general

Author Response

Thank you so much for your kind and constructive comments. please see the attachment.

Reviewer 3 Report

Journal: biomolecules

Manuscript ID: biomolecules-2138175

Title of the Manuscript: Recent Developments in Biopolymer-Based Hydrogels for Tissue Engineering Applications

The review article describes biopolymer-based hydrogels for issue engineering applications, DDS systems and wound dressings are limited. As over all the manuscript is well designed; results and discussions are well adequate. Hence the manuscript is recommended for publication in the journal ‘biomolecules’. However, authors need to fulfil the below comments

         Authors need to discuss in specific about the functionality & structure of biopolymers their significance in tissue engineering

Author Response

Thank you so much for your kind comment. Please see the attachment.
